# Cross-National Study of Worrying, Loneliness, and Mental Health during the COVID-19 Pandemic: A Comparison between Individuals with and without Infection in the Family

**DOI:** 10.3390/healthcare9070903

**Published:** 2021-07-16

**Authors:** Tore Bonsaksen, Janni Leung, Mariyana Schoultz, Hilde Thygesen, Daicia Price, Mary Ruffolo, Amy Østertun Geirdal

**Affiliations:** 1Department of Health and Nursing Sciences, Faculty of Social and Health Sciences, Inland University of Applied Sciences, Hamarvegen 112, 2418 Elverum, Norway; 2Faculty of Health Studies, VID Specialized University, 4306 Sandnes, Norway; 3Faculty of Health and Behavioural Science, The University of Queensland, St. Lucia, QLD 4072, Australia; j.leung1@uq.edu.au; 4Faculty of Health and Life Sciences, Northumbria University, Newcastle upon Tyne NE1 8ST, UK; mariyana.schoultz@northumbria.ac.uk; 5Department of Occupational Therapy, Prosthetics and Orthotics, Faculty of Health Sciences, Oslo Metropolitan University, 0130 Oslo, Norway; hilde.thygesen@oslomet.no; 6Faculty of Health Studies, VID Specialized University, 0370 Oslo, Norway; 7School of Social Work, University of Michigan, Ann Arbor, MI 48109, USA; daiciars@umich.edu (D.P.); mruffolo@umich.edu (M.R.); 8Department of Social Work, Faculty of Social Sciences, Oslo Metropolitan University, 0130 Oslo, Norway; amyoge@oslomet.no

**Keywords:** COVID-19, coronavirus, cross-national study, loneliness, mental health, pandemic, social distancing, worry

## Abstract

Objective: The objective of this study was to examine differences in worry, loneliness, and mental health between those individuals infected by COVID-19 or having someone their family infected, and the rest of the population. Methods: A cross-sectional online survey was conducted in Norway, UK, USA, and Australia during April/May 2020. Participants (*n* = 3810) were recruited via social media postings by the researchers and the involved universities. Differences between those with and without infection in the family were investigated with chi-square tests and independent *t*-tests. Multiple regression analyses were used to assess associations between sociodemographic variables and psychological outcomes (worry, loneliness, and mental health) in both groups. Results: Compared to their counterparts, participants with infection in the family reported higher levels of worries about themselves (*p* < 0.05) and their family members (*p* < 0.001) and had poorer mental health (*p* < 0.05). However, the effect sizes related to the differences were small. The largest effect (*d* = 0.24) concerned worries about their immediate family. Poorer psychological outcomes were observed in those who were younger, female, unemployed, living alone and had lower levels of education, yet with small effect sizes. Conclusions: In view of the small differences between those with and without infection, we generally conclude that the mental health effects of the COVID-19 situation are not limited to those who have been infected or have had an infection within the family but extend to the wider population.

## 1. Introduction

As the COVID-19 pandemic became a reality in Western countries in the beginning of March 2020, strict national policies regarding public behavior were implemented in countries throughout the world, including Europe, America, and Australia. In conjunction with hygiene rules, social distancing became the main policy for public behavior [1,2]. The policy implied having minimal contact with persons from outside the household, and people were generally encouraged to stay at home. Nurseries, schools, and universities were closed, and practically overnight, working from home and attending online classes became the new standard for workers and students. Flights and travels were cancelled, as were all sports, religious, and cultural events. Non-vital businesses requiring physical proximity, such as hairdressers and physiotherapy clinics, were closed. Consequently, many businesses experienced financial problems, and many employees were temporarily furloughed from their jobs [3]. As higher age and chronic disease have been associated with increased risk of experiencing severe outcomes from contracting COVID-19 infection [4], individuals in these groups were instructed to be particularly careful regarding exposure to other people.

While studies of medical treatments, drugs and vaccines were developed rapidly after the pandemic outbreak, studies concerned with the psychological and social consequences of the global response to the pandemic are of no less importance. Social distancing, especially the more severe forms such as social isolation and quarantine without the possibility of meeting other people, can result in high levels of psychological distress and even post-traumatic stress disorder [5,6]. Social isolation has also been associated with increased risks of chronic disease and premature mortality [7]. In a study conducted during the first phase of the COVID-19 outbreak in China, more than half of the respondents rated the psychological impact as moderate-to-severe, and about one-third reported moderate-to-severe anxiety [8]. However, access to specific and up to date health information and using precautionary measures were associated with lower levels of stress, anxiety, and depression.

From a global perspective, isolation and social distancing have been included among a range of factors that may aggravate the longer-term impact of COVID-19 on mental health [9,10]. On the other hand, previous research has demonstrated that adequate disease knowledge (health literacy) and social contact and support are strong predictors of health outcomes [11,12]. Social contact and support serve as a ‘buffer’ between stress and mental health problems [13]. This is not only when support is received but also when support is given—the findings from a Japanese study showed that men under stressful circumstances were less depressed when they received or provided social support [14]. Thus, the social distancing measures taken to avoid the spread of COVID-19 appear to have the potential to undermine people’s mental health and sense of belonging in the community.

Physical diseases have frequently been shown to be associated with poorer mental health, among young [15] and older people alike [16]. Thus, individuals infected with COVID-19 may also experience reduced mental health, although with substantial individual variation. However, in view of the everyday life constraints that people have experienced during a time of social distancing, mental health may not only be reduced among those who have been infected by the disease or have witnessed family members or close ones fall ill. The hypothesis driving this study is that the social distancing context of COVID-19 affects the public mental health, and not only the mental health of those who have been directly exposed to the disease. To that effect, the social media coverage of the COVID-19 pandemic, often showcasing the worst possible effects of the disease (e.g., people receiving intensive care in hospitals, alarming death rates), has been enormous. A recent study from China found that participants with more exposure to social media had higher odds of anxiety, alone and in combination with depression [17]. General population studies comparing mental health measures of those directly affected by COVID-19 with those who were not, are lacking. This study contributes to filling this gap in the literature. If the study hypothesis is supported, mental health policies in the pandemic context would need to target the general population, and not only those who have been most directly affected. Therefore, the aim of this study was to examine the differences in worry, loneliness, and mental health between those individuals infected by COVID-19 or having someone their family infected, and the rest of the population. The specific research question for the study was: Are people who have been infected with the coronavirus, or have had family members infected, more worried, lonely, and mentally distressed compared to those without this experience with COVID-19 infection?

## 2. Methods

### 2.1. Setting

The researchers behind the study were based in four different countries: Norway, USA, UK, and Australia. Therefore, the general population in these countries were invited in April/May 2020 to participate in a self-administered survey that was distributed via different social media, such as Facebook, Instagram, and Twitter. Each participating country had a landing site for the survey at the involved universities. These were Oslo Metropolitan University, Norway; University of Michigan, USA; University of Salford, UK; and University of Queensland, Australia. A.Ø.G. from Oslo Metropolitan University initiated the overall project, but each of the universities had a project lead. The survey was translated from Norwegian to English by the researchers according to the language and cultural context where the survey was to be used. All data used in the study are based on responses to the online survey. 

### 2.2. Participants

A convenience sample was recruited through the use of various social media linking to the online survey (outlined above). To be included in the study, participants were required to be 18 years or older; to understand the language in which the survey was presented (Norwegian or English), and to be living in one of the relevant countries at the time of the survey (Norway, USA, UK or Australia).

### 2.3. Measures

#### 2.3.1. Sociodemographic Characteristics

Sociodemographic variables included age group (18–29 years, 30–39 years, 40–49 years, 50–59 years, 60–69 years, 70 years and above), gender (male, female, other/not stated), living area (rural/farming area, small town, medium-sized city, large city), highest completed education level (elementary school, high school, associated/technical degree, bachelor’s degree, master’s/doctoral degree), cohabitation (living with spouse or partner), and employment status (having full-time or part-time employment versus not having employment).

#### 2.3.2. Worry

Three aspects of worry were measured. The participants were asked: (i) to what extent are you worried about your own situation, (ii) to what extent are you worried about your immediate family, and (iii) to what extent are you worried about the future? All items had the following response options: (1) not at all, (2) slightly worried, (3) worried, (4) very worried, and (5) overwhelmed. The items were constructed for this study and analysed as individual items. 

#### 2.3.3. Loneliness

Loneliness was measured with the Loneliness Scale [18] which consists of six statements, all of which rated from 0 (totally disagree) to 4 (totally agree). This scale was designed to measure two different aspects of loneliness, “emotional loneliness” and “social loneliness”. Previous factor-analytic studies have found the six statements to load on two different factors, and that they, therefore, should be treated as constituting two different scales reflecting the two different aspects of loneliness [18,19]. Internal consistencies (Cronbach’s α) in this study were 0.66 and 0.86 for the emotional loneliness and social loneliness scales, respectively.

#### 2.3.4. Mental Health

General Health Questionnaire 12 (GHQ-12) is widely used as a self-report measure of mental health [20,21]. A large number of studies in the general adult, clinical, work and student population have provided support for its validity across samples and contexts [22,23,24,25,26]. Six items of the GHQ-12 are phrased positively (e.g., ‘able to enjoy day-to-day activities’), while six items are phrased as a negative experience (e.g., ‘felt constantly under strain’). For each item, the person indicates the degree to which the item content has been experienced during the two preceding weeks, using four response categories (‘less than usual’, ‘as usual’, ‘more than usual’ or ‘much more than usual’). Items are scored between 0 and 3, and positively formulated items are recoded prior to analysis. As a result, the GHQ-12 scale score range is 0–36, with higher scores indicating poorer mental health (more psychological distress). Cronbach’s *α* for the GHQ-12 was 0.87.

#### 2.3.5. COVID-19 Infection

Participants were asked two questions relating to COVID-19 infection: (i) have you been infected by COVID-19; and (ii) has someone in your immediate family been infected by COVID-19? Both questions were answered with ‘yes’, ‘no’ or ‘don’t know’. In view of the small number of participants who reported COVID-19 infection, we compared those who had been personally infected with COVID-19 (*n* = 52, 1.4%) with those reporting that someone in the immediate family had been infected (*n* = 373, 9.8%). Between these groups, the difference regarding social loneliness (*M* = 4.9 versus *M* = 3.9, *p* < 0.05) was the only difference reaching statistical significance. Therefore, we grouped those individuals personally infected with individuals who had a family member infected and with those reporting both, into one category (*n* = 504, 13.2%). The rest of the participants (*n* = 3306, 86.8%) constituted the comparison group (those indicating not known infection or no infection, either personally or within the family). For the remainder of this article, we will label those participants with infection personally or within the immediate family as ‘with infection’, while the rest of the participants will be labelled ‘without infection’ in the family.

### 2.4. Statistical Analysis

The overall sample, and each of the national subsamples, were described with frequencies and percentages for categorical variables and means and standard deviations for continuous variables. Overall differences in proportions between groups were analyzed with the chi-square test.

Methodology studies have found that deviations from the normal distribution is common and often unproblematic in large public health datasets [27], and that parametric statistical tests can be used in large studies instead of non-parametric tests, even in cases of heavily skewed distributions [27], given that conditions for their use are met. In our study, the outcome variables’ distribution deviated from the normal distribution (all Kolmogorov-Smirnov tests *p* < 0.001) but was not heavily skewed (skewness < 0.80 for all). When comparing the groups using both parametric and non-parametric methods, the results were identical across methods. Thus, we proceeded with the use of parametric statistical tests. Differences in worry, loneliness, and mental health between those with and without infection were analyzed with independent *t*-tests for the whole sample and for each of the four countries. A series of linear regression analyses, stratified on infection status, was used to assess associations between sociodemographic variables and a range of outcomes: worry for oneself, worry for family members, worry about the future, emotional loneliness, social loneliness, and mental health. In each of the regression analyses, included independent variables were age group, gender, education level, employment and living with spouse or partner. Statistical significance was set at *p* < 0.05. Missing values were managed with casewise deletion, resulting in *n* varying between analyses.

### 2.5. Ethics

The data in this cross-sectional cross-national study were collected anonymously. The study was quality assured and approved by Oslo Metropolitan University and by the Regional Committee for medical and health research ethics (REK; project reference 132066) in Norway. In the USA, it was reviewed by the University of Michigan Institutional Review Board for Health Sciences and Behavioral Sciences (IRB HSBS) and designated as exempt (HUM00180296). Similarly, it was reviewed by the University Health Research Ethics (HSR1920-080) in the UK and the University of Queensland Human Research Ethics Office (HSR1920-080; 2020000956) in Australia.

## 3. Results

### 3.1. Participants

The sociodemographic characteristics of the sample subgroups are displayed in Table 1. Overall, the distribution of sociodemographic characteristics was similar between those with and without infection. Fifty-nine percent of the sample was below the age of 50 years, and 74% had a bachelor’s degree education or higher. Seventy-one percent were in employment, and 85% reported living with a spouse or partner. There was a larger proportion of women who were classified as ‘with infection’, compared to men. 

### 3.2. Differences between Participants with and without Infection

Table 2 displays the results from the independent *t*-tests of differences in the whole sample between those with and without infection. Compared to their counterparts, participants with infection in the family rated that they were significantly more worried about their own situation and about family members, and they had poorer mental health. However, the effect sizes related to the differences were small, with the largest effect (*d* = 0.24) concerned with the difference in worry about the immediate family. 

The proportion of participants exposed to infection differed between countries (*p* < 0.001). Norway had 124 individuals with infection (16.1% of the 771 participants from Norway), while the UK had 224 (16.3% of the participants from the UK), USA had 140 (10.1% of the participants from USA) and Australia had 16 (5.9% of the participants from Australia). Rerunning the analyses by country, the main pattern of small to negligible differences was retained, with some minor divergence from the overall results. In Norway, participants with infection (self or within the immediate family) were more worried about their family members (*p* < 0.05) and felt more socially lonely (*p* < 0.05) than their counterparts without infection. In the UK, participants with infection were more worried about their family members (*p* < 0.01) than their counterparts. In the USA, no group differences occurred on any of the employed variables. In Australia, the group with infection was too small for meaningful comparison against their counterparts.

### 3.3. Factors Associated with Worrying, Loneliness, and Mental Health

Among those with infection (self or in the family), higher age was associated with lower emotional loneliness and better mental health. Female gender was associated with poorer mental health. Being employed was associated with less worry about the future and lower emotional loneliness, while living with a spouse or partner was associated with lower emotional and social loneliness (see Table 3).

Among those without experience of infection, higher age was associated with more worry about one’s own situation, less worry about the future, lower emotional loneliness and better mental health. Compared to men, women experienced more worry in all domains, and they experienced more emotional loneliness, less social loneliness and poorer mental health. Having higher education, having employment and living with a spouse or partner were all associated with less worry in all domains, lower emotional and social loneliness, and better mental health (see Table 4).

## 4. Discussion

This study aimed to examine the differences in worries, loneliness, and mental health between those with and without experience of COVID-19 infection personally or in the immediate family. Despite significantly higher levels of worry and poorer mental health among those experiencing infection, the effect sizes associated with the results demonstrate that differences between those with and without infection experience were small to negligible. The associations between sociodemographic variables and worry, loneliness, and mental health were similar for the two groups, implying that the psychological impacts of COVID-19 may be extended across the population, even among those without infection. 

In comparison to previous studies using the GHQ to measure mental health in general populations [25,26], as expected, this study sample had much poorer mental health. The main finding of this study is that the differences between those with and without infection themselves or in the immediate family were small to negligible. This supports the hypothesis that the COVID-19 situation affects the mental health in the general population, and not just the mental health of those who are most directly affected. This interpretation is also supported by other studies of COVID-19 [9,10], suggesting that the COVID-19 outbreak may have adverse psychological impacts beyond the individual, extending to community and global levels. 

While statistically significant differences were found regarding worry about own situation, worry about family, and mental health, effect sizes were generally small. The large number of individuals in our sample contributed toward making even very small effect sizes reach statistical significance. The largest effect was found for worry about family members, where those with infection were more worried than those without infection. It is understandable that those with infection felt worried about other family members potentially contracting the disease or worry about family members who had already been infected. In particular, having older family members or family members with underlying chronic disease (cardiovascular disease, chronic obstructive pulmonary disease, hypertension, diabetes and cerebrovascular disease) may have increased worry, given that these diseases are known and major risk factors for poor outcomes among patients with COVID-19 infection [4]. Given that the larger proportion of those ‘with infection’ consisted of individuals reporting infection in the immediate family, the explanation emphasizing worry for family members who had contracted the disease seems viable in most cases. Rerunning the analyses for each of the countries, the main pattern from the analysis of the whole sample was retained, despite some minor discrepancies and varied baseline levels of infection rates.

For the group without infection in the immediate family, most associations were weak but statistically significant. This is indicative of a high-powered study to detect even very small effects in the data [28]. However, the associations were in the same direction (or were near zero) in both groups of participants. Focusing on associations with effect sizes above 0.10, the study showed that higher age was associated with lower emotional loneliness and better mental health in both groups. This was somewhat a surprise finding because older people are at higher risk from COVID-19, so we may have expected higher levels of worry in this age group. However, our finding is mostly in line with previous studies, in which higher age has been associated with less anxiety [29] and less depression [30], but also with more loneliness [31]. However, the age-loneliness association has been found to vary between countries [32]. Further, the association was specifically concerned with emotional loneliness (e.g., feeling rejected), while the association with social loneliness (e.g., having no one to turn if needed) was near zero. The notion of better mental health among those with higher age warrants further investigation. For example, different types of anxiety disorders have been found to have different prevalence across age groups, with phobias having the highest prevalence among children and adolescents, while panic disorder and PTSD are most prevalent in adulthood, and worry (i.e., generalized anxiety disorder) is most prevalent among older adults [32]. While social distancing may affect people similarly across age groups with regard to their possibility of maintaining social interactions, the emotional consequences of reduced social interaction may be more outspoken among individuals of younger age. Older and younger adults may also use social media for different reasons and with different frequency, which may have psychological impacts. Social distancing policies may also affect older adults differently depending on their retirement and residential status. 

We found that women had poorer mental health than men. This is in line with a vast amount of research—women have been found to be more susceptible to most types of mental health problems except alcohol and drug misuse, which is more common among men [33,34,35,36]. Thus, it appears the COVID-19 situation may have perpetuated the gender gap with respect to mental health. 

Among those with infection in the family, having higher education was not significantly associated with any of the psychological outcomes. In those without infection, having higher education was weakly associated with most outcomes. Perhaps of most interest, though, there were no significant differences in worry for family members between participants with and without higher education. Thus, while higher education, by increasing knowledge and self-efficacy, may be a resource for reducing worry about one’s own situation and worry about the future, one’s worry about family members appears not to vary by education level. This finding adds nuance to the general conception that higher levels of education are associated with better health [37]. Our findings imply that education may not buffer against the worries instigated by the possibility of having family members exposed to an ongoing virus pandemic. If having relevant knowledge about the disease and self-efficacy related to implementing measures of prevention are crucial mechanisms in determining health outcomes [38], the results make sense. While the control over one’s own behavior and exposure to risk is substantial, the control one can impose on others’ behavior and exposure to risk is minimal.

Having employment was associated with less worry about the future and less loneliness. Among those without infection, employment was also associated with better mental health. During the first phase of the COVID-19 outbreak, many businesses were closed and employees were temporarily furloughed [3]. Given the importance of employment for income and social interactions, it is understandable that those who were employed were less worried and less lonely than their counterparts who were unemployed. Although it might be preferable to conceptualize employment as a continuum rather than a categorical measure [39], having employment per se has previously been shown to be related to lower odds of depression [30] and better global health [40] in a general population sample. 

The association between living with spouse/partner and reporting less loneliness and better mental health is similarly understandable. The paired relationship is often the basis for experiencing regular social contact and support, which in turn has been found to be a strong predictor of health outcomes [11]. Not only does the continued affection and support from another person counteract potential effects of stress on mental health [13], the paired relationship also allows for experiencing the benefits of providing support to the other person [14].

## 5. Study Limitations

The study is limited in several ways. The data were collected using a cross-sectional online survey, therefore assumptions about causal relationships should not be made. We do not know how well the sample is representative of the population of people who used social media in the four respective countries. Our study sample had a higher proportion of female, well-educated and urban participants. However, the age distributions were generally considered well in accordance with general population statistics. On the other hand, response to the general population targeted advertisement in Australia was low, resulting in a large proportion of participants being recruited among followers of the university’s social media postings. Thus, among the Australian participants, there was an over-representation of younger participants with university degrees. However, the sampling of participants from four different countries reduces the risk of severe sampling bias [41].

The sample was recruited through advertisements released by the university through social media and by personal postings and shares on social media. Thus, the results may not be generalized beyond those who use social media relatively frequently. The degree of disease outbreak and social distancing policies differed between states within the USA, which warrants deeper investigation. The internal consistency of the emotional loneliness scale was lower than the recommended 0.70 threshold. However, lower internal consistency estimates are common for shorter scales [42,43]. This consideration applies to the three-item emotional loneliness scale. Subpopulations, such as older adults and those who have experienced job loss, should be examined in future research.

## 6. Conclusions

Previous studies concerned with responses to the COVID-19 pandemic have frequently targeted specific groups of interest, whereas studies comparing responses between the general population and those most closely affected—by being infected by the virus themselves, or by having family members infected—appear to be missing. Therefore, this study contributes to the literature by examining differences in worries, loneliness, and mental health between individuals infected by COVID-19 or having someone in their family infected, and the rest of the population. The results of the study showed that those in the group exposed to infection reported more worry about self and the immediate family, and worse mental health, but the differences were small. Thus, the specific contribution of this study lies in its demonstration of mental health effects of the COVID-19 situation extending to the wider population—they are not limited to those who have been infected or have experienced someone with infection in the family. This implies that mental health policies during and following the pandemic need to target the general population as a whole, and not only those who have been most directly affected.

## Figures and Tables

**Table 1 healthcare-09-00903-t001:** Sociodemographic characteristics among individuals with and without COVID-19 infection.

	Without Infection (*n* = 3306, 86.8%)		With Infection (*n* = 504, 13.2%)		
*Characteristics*	*n*	%	*n*	%	*p*
Age group					0.56
18–29 years	619	87.8	86	12.2	
30–39 years	628	88.1	85	11.9	
40–49 years	721	87.2	106	12.2	
50–59 years	617	85.3	106	14.7	
60–69 years	526	85.9	86	14.1	
70+ years	191	85.3	33	14.7	
Gender					<0.01
Male	647	90.1	71	9.9	
Female	2607	85.9	427	14.1	
Living area					0.35
Rural/farming	244	86.5	38	13.5	
Small town	736	87.3	107	12.7	
Medium-sized city	1079	87.9	149	12.1	
Large city	1247	85.6	210	14.4	
Education level					0.27
Lower education	848	85.7	141	14.3	
Bachelor’s degree or higher education	2457	87.1	363	12.9	
Living with spouse/partner					0.93
Yes	2020	86.4	319	13.6	
No	1041	86.5	163	13.5	
Employment					
Yes, full-time or part-time	2340	86.9	352	13.1	0.72
No	966	86.5	151	13.5	

*Note.* Statistical tests are chi-square tests. Proportions are within categories.

**Table 2 healthcare-09-00903-t002:** Worry, loneliness, and mental health among participants with and without COVID-19 infection.

Variables	Without Infection	With Infection	Difference	Effect Size	
*Worry*	*M* (*SD*)	*M* (*SD*)	*M* (*SD*)	Cohen’s *d*	*p*
Worry about own situation	2.38 (1.03)	2.52 (1.11)	0.14	0.13	<0.05
Worry about immediate family	2.75 (1.00)	2.99 (1.03)	0.25	0.24	<0.001
Worry about the future	2.75 (1.11)	2.83 (1.06)	0.09	0.07	0.16
*Loneliness*					
Emotional loneliness	6.01 (2.69)	6.18 (2.65)	0.17	0.06	0.18
Social loneliness	3.89 (2.98)	4.12 (3.19)	0.22	0.08	0.14
*Mental health*					
GHQ score	16.19 (6.87)	17.09 (7.72)	0.90	0.13	<0.05

*Note*. Statistical tests are independent *t*-tests.

**Table 3 healthcare-09-00903-t003:** Standardized *β* weights of adjusted associations between sociodemographic factors and worrying, loneliness, and mental health among participants with infection (*n* = 504).

Sociodemographic Variables	Worry Self	Worry Family	Worry Future	Emotional Loneliness	Social Loneliness	Mental Health
Age group	0.02	0.08	−0.05	−0.19 ***	−0.06	−0.13 **
Gender	0.10	−0.04	−0.02	0.04	−0.04	0.09 *
Education	−0.08	−0.09	−0.08	−0.07	−0.07	−0.07
Employment	−0.09	−0.03	−0.15 **	−0.11 *	−0.08	−0.09
Living with spouse/partner	−0.06	−0.11	−0.07	−0.13 **	−0.16 **	−0.05
**Explained variance**	**3.2%**	**2.8%**	**4.0% ***	**7.8% *****	**4.4% ****	**4.5% ****

*Note*. Table content is standardized *β* weights, indicating the strength of the association between the sociodemographic variables and the dependent variables (worrying, loneliness, and mental health) while adjusting for all included sociodemographic variables. Variable coding: higher age group is higher age; higher gender is female; higher education is having bachelor’s degree education or higher; higher employment is having full-time or part-time employment (as opposed to not having employment); living with spouse/partner is 1, while not living with spouse/partner is 0. Higher scores on worry and loneliness indicate higher levels, whereas higher ratings on mental health indicate poorer mental health. * *p* < 0.05, ** *p* < 0.01, *** *p* < 0.001.

**Table 4 healthcare-09-00903-t004:** Standardized *β* weights of adjusted associations between sociodemographic factors and worrying, loneliness and mental health among participants without infection (*n* = 3306).

Independent Variables	Worry Self	Worry Family	Worry Future	Emotional Loneliness	Social Loneliness	Mental Health
Age group	0.05 *	−0.02	−0.05*	−0.19 ***	−0.00	−0.16 ***
Gender	0.07 **	0.08 ***	0.08 ***	0.04 *	−0.05 **	0.10 ***
Education	−0.10 ***	−0.04	−0.09 ***	−0.08 ***	−0.06 ***	−0.07 ***
Employment	−0.09 ***	−0.05*	−0.16 ***	−0.09 ***	−0.10 ***	−0.10 ***
Living with spouse/partner	−0.06 **	−0.05*	−0.08 ***	−0.18 ***	−0.17 ***	−0.13 ***
**Explained variance**	**3.6% *****	**1.4% *****	**5.7% *****	**10.4% *****	**4.9% *****	**7.1% *****

*Note*. Table content is standardized *β* weights, indicating the strength of the association between the sociodemographic variables and the dependent variables (worry, loneliness, and mental health) while adjusting for all included sociodemographic variables. Variable coding: higher age group is higher age; higher gender is female; higher education is having bachelor’s degree education or higher; higher employment is having full-time or part-time employment (as opposed to not having employment). Living with spouse/partner is 1, while not living with spouse/partner is 0. Higher scores on worry and loneliness indicate higher levels, whereas higher ratings on mental health indicate poorer mental health. * *p* < 0.05, ** *p* < 0.01, *** *p* < 0.001.

## Data Availability

The dataset analyzed for this study will be available from Oslo Metropolitan University on request after the completion of the study. E-mail amyoge@oslomet.no.

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
