# Peer review of "Cross-National Study of Worrying, Loneliness, and Mental Health during the COVID-19 Pandemic: A Comparison between Individuals with and without Infection in the Family"

_healthcare, 2021, doi:10.3390/healthcare9070903_

Round 1
Reviewer 1 Report
First of all, I am grateful for the opportunity to review this paper. COVID-19 is an ongoing pandemic that has resulted in global health, economic and social crises. Currently, few evidences concerning knowledge, attitudes, and practices related to COVID-19 pandemic are reported in the scientific literature. Moreover, addressing the psychological effects of infectious diseases outbreak is challenging, since most efforts and services are intended to protect physical wellbeing and less attention is paid to the psychological side. In this context, the research under review is aimed to examine differences in worry, loneliness, and mental health between those individuals infected by COVID-19 or having someone their family infected, and the rest of the population.
The article is interesting and may provide interesting information for public health. Nevertheless, it must be improved:
Introduction: The authors should make it clear about what is the gap in the literature that is filled with this study? First of all what are level of knowledge related to COVID-19 among other Countries (refer to Gallè F et al. Awareness and Behaviors Regarding COVID-19 among Albanian Undergraduates. Behav Sci (Basel). 2021 Mar 31;11(4):45. doi: 10.3390/bs11040045) What is the contribution of the study to the literature? What are the implications of the study?
Methods: source of data and sampling procedure will benefit from more detail. Why did the authors choose Norway, USA, UK and Australia? What is the referring population and how did the authors evaluated if the sample is representative? 3000 questionnaires are a big number but not too much if compared to the number of enrolled countries. What about validation of the questionnaire? Face validity, intelligibility?
Discussion: I also suggest expanding. Emphasize the contribution of the study to the literature, the implications and recommendations based on previous experience. The authors should compare the results with those of longer follow up studies (refer to Roma P, et al. A 2-Month Follow-Up Study of Psychological Distress among Italian People during the COVID-19 Lockdown. Int J Environ Res Public Health. 2020 Nov 5;17(21):8180. Limits section must be improved.
Reviewer 2 Report
This reviewer commends the authors for their study.
The research question(s) is/ are not well defined and it is not stated how research fills an identified knowledge gap.
What is the research question?
What is the knowledge gap being investigated?
How does this research contribute to filling that gap?
The absence of a sampling frame and the inherent bias in convenience sampling and thus inability to make generalizations from this study should be mentioned in the limitations.
Round 2
Reviewer 1 Report
The paper was partially modified, but in my opinion not sufficiently to make it acceptable for pubblication. Great concern is from a methodological point of view. The sample seems not adeguately choosen and irrationale: four countires (Norway, USA, UK and Australia) with no apparent link were enrolled for a total of more than 3000 questionnaire (not proportional or representative of the target population). This may biased the results. No precise information are reported with regard to the validation process of part of the used questionnaire (the reply is: "we do not have information about the validity of the questions that were specifically developed for this study".)
Furthermore, level of knowledge regarding the disease may impact on worry, loneliness, and mental health, therefore must be cited, at least in the introduction.
